# Zero-Fluoroscopy Catheter Ablation of Supraventricular Tachycardias in the Pediatric Population

**DOI:** 10.3390/children10091513

**Published:** 2023-09-06

**Authors:** Mirko Topalović, Matevž Jan, Tine Prolič Kalinšek, David Žižek, Jernej Štublar, Rina Rus, Dimitrij Kuhelj

**Affiliations:** 1Cardiology Department, Pediatric Clinic, University Medical Centre Ljubljana, Bohoriceva 20, 1000 Ljubljana, Slovenia; 2Cardiovascular Surgery Department, Surgical Clinic, University Medical Centre Ljubljana, Zaloska 7, 1000 Ljubljana, Sloveniatine.prolic.kalinsek@kclj.si (T.P.K.); jernej.stublar@kclj.si (J.Š.); 3Cardiology Department, Internal Medicine Clinic, University Medical Centre Ljubljana, Zaloska 7, 1000 Ljubljana, Slovenia; david.zizek@kclj.si; 4Clinical Institute of Radiology, University Medical Centre Ljubljana, Zaloska 7, 1000 Ljubljana, Slovenia; dimitrij.kuhelj@kclj.si

**Keywords:** zero-fluoroscopy, catheter ablation, supraventricular tachycardia, children

## Abstract

Catheter ablation (CA) of supraventricular tachycardias (SVTs) is conventionally performed with the aid of X-ray fluoroscopy. Usage of a three-dimensional (3D) electro-anatomical mapping (EAM) system and intracardiac echocardiography (ICE) enables zero-fluoroscopy ablation, eliminating the harmful effects of radiation. We retrospectively analyzed the feasibility, effectiveness and safety of zero-fluoroscopy radiofrequency and cryoablation of various types of SVTs in pediatric patients. Overall, in 171 consecutive patients (12.5 ± 3.9 years), 175 SVTs were diagnosed and 201 procedures were performed. The procedural success rate was 98% (193/197), or more precisely, 100% (86/86) for AVNRT, 95.8% (91/95) for AVRT, 94.1% (16/17) for AT and 100% (2/2) for AFL. No complications were recorded. Follow-up was complete in 100% (171/171) of patients. During the mean follow-up period of 488.4 ± 409.5 days, 98.2% of patients were arrhythmia-free with long-term success rates of 98.7% (78/79), 97.5% (78/80), 100% (13/13) and 100% (2/2) for AVNRT, AVRT, AT and AFL, respectively. Zero-fluoroscopy CA of various types of SVTs in the pediatric population is a feasible, effective and safe treatment option.

## 1. Introduction

Supraventricular tachycardias (SVTs) are the most common arrhythmias in the pediatric population. The prevalence of SVTs is reported to be 2.25/1000 persons in the general population, with an annual incidence in children of 13/100,000 person-years [1,2]. The majority of cases of SVTs are atrioventricular nodal reentry tachycardias (AVNRTs) and atrioventricular reentry tachycardias (AVRTs), while atrial tachycardia (AT) is responsible for about 10% of cases. The prevalence of each tachycardia also varies depending on the age. While AVRT is more common in infancy and in young children, the prevalence of AVNRT is higher later in the teenage years.

The mainstay treatment for SVT is catheter ablation (CA) [3]. Traditionally, CA is performed with the aid of fluoroscopy. However, fluoroscopy exposes the patient and laboratory staff to the harmful effects of ionizing radiation. It has been postulated that these effects are especially harmful to children as they are more radiosensitive. The reasons for this may be their more rapidly dividing cells and increased life expectancy, thus increasing the latency period for a malignancy to develop [4]. In addition, operators’ exposure to ionizing radiation, due to their close proximity to the source and the scatter effect, is also of concern [5,6]. Moreover, wearing heavy radioprotective aprons has been associated with the development of orthopedic diseases, with spinal injuries being the most common [7]. The importance of the reduction and elimination of fluoroscopy has been recognized by the International Commission on Radiation Protection with the inclusion of the ALARA (as low as reasonably achievable) principle in its guidelines for pediatric interventional procedures [8].

In recent years, the development of non-fluoroscopic imaging modalities, such as three-dimensional electroanatomical mapping (3D EAM) systems and intracardiac echocardiography (ICE), has enabled operators to reduce or completely eliminate the need for fluoroscopy during CA of various SVTs. Yang et al. [9] showed in a meta-analysis of zero and near-zero fluoroscopy-guided CA of various arrhythmias that the reduction and/or elimination of fluoroscopy is feasible, safe and effective. However, the meta-analysis included only studies with adult patients. These findings were confirmed with the more recent and larger meta-analysis published in 2022 by Debreceni et al. [10], which included six pediatric studies. Studies investigating a reduction in fluoroscopy in pediatric patients usually include the use of fluoroscopy for the transseptal puncture (TSP) or include only right-sided SVTs in their analysis [11,12,13,14,15,16,17,18,19,20,21,22]. To overcome the issue of the use of fluoroscopy for the left-sided approach, investigators used transesophageal echocardiography (TOE) for the TSP, utilized an arterial transfemoral retrograde approach, or used a patent foramen ovalis [20,21,22]. Recently, Žižek et al. [23] showed in a single-center retrospective study that TSP can be safely performed under ICE guidance even in pediatric patients.

The aim of this study was to investigate the feasibility, efficacy and safety of the zero-fluoroscopy approach with the aid of the 3D EAM system and ICE in pediatric patients with right- and left-sided SVTs.

## 2. Materials and Methods

### 2.1. Study Population

Our retrospective analysis included 171 consecutive pediatric patients referred to our institution for CA of SVTs from April 2014 to October 2021. Written informed consent to undergo the CA was obtained from all patients, their parents or legal guardians. All patients underwent a pre-procedural clinical examination, routine blood biochemistry laboratory tests and pre-procedural echocardiography. All antiarrhythmic drug (AAD) therapy was discontinued before the procedures.

### 2.2. Electrophysiology Study

Procedures were performed under general anesthesia in patients younger than 14 years, while local anesthesia and conscious sedation were used in older patients. Femoral vein access was obtained under ultrasound guidance. The 3D EAM system (EnSite NavX, Ensite Velocity, Ensite Precision, Abbott, St. Paul, MN, USA or Carto 3, Biosense Webster, Diamond Bar, CA, USA) was used for guidance of the catheters in the heart and advancement through the vasculature.

After femoral vein access was obtained, a 10-polar diagnostic catheter was advanced into the right atrium. The catheter was used to construct a partial 3D rendering of the right atrium and to mark the location of His potential. The catheter was then placed in the coronary sinus. Next, an additional 4- or 10-polar diagnostic catheter was inserted into the heart and placed on the basal section of the right side of the interventricular septum.

A standard electrophysiology study followed, with the aim of tachycardia induction. In cases of clear ventricular preexcitation, the induction of tachycardia was left to the physician’s discretion. If induction of tachycardia was not achieved or conduction over the accessory pathway (AP) was not detected, the protocol was repeated with an isoprenaline challenge. Standard diagnostic maneuvers were employed as needed to determine the type of induced tachycardia.

### 2.3. Left-Sided Access

Left-sided access was obtained with ICE (AcuNav, Siemens Healthineers AG, Erlangen, Germany)-guided TSP in all patients.

ICE was further used for the navigation of catheters in the heart at the physician’s discretion.

### 2.4. Mapping and Ablation of AVNRT

The right inferior extension of the AV node was targeted. A non-irrigated radiofrequency ablation (RFA) catheter or a 4 mm or 6 mm tip cryoablation (CRA) catheter (Freezor and Freezor Xtra, Medtronic, Minneapolis, MN, USA) was used to terminate the conduction over the right inferior extension of the AV node. RFA catheters were used in a temperature-controlled fashion with power titration from 20 to 40 W, with the goal to achieve junctional rhythm during the ablation. When CRA catheters were used, cryo mapping (−30 °C) was performed during ongoing tachycardia or during programmed atrial stimulation with manifested conduction over the slow pathway. If the tachycardia was terminated during cryo mapping, or the conduction over the slow pathway was terminated, the cryo mapping was then switched to CRA (−80 °C), usually for 240 s. At least one additional lesion was applied in close proximity to the successful one. If the tachycardia was mechanically terminated when a CRA catheter was used, a lesion was applied at the spot of mechanical termination when possible.

### 2.5. Mapping and Ablation of AVRT

Mapping of the earliest atrial or ventricular potential and a search for the accessory pathway (AP) potential on the tricuspid or mitral annulus were performed either during ongoing orthodromic AVRT, during ventricular pacing, or during ventricular preexcitation in sinus rhythm. Irrigated RFA catheters (D curve FlexAbilityTM, Abbott, Abbott Park, IL, USA or D curve Celsius^®^ Biosense Webster, Irvine, CA, USA or ThermoCool SmartTouch SF Biosense Webster, Irvine, CA, USA) were used in a temperature-controlled fashion for the ablation of APs in the right or left atrium. Contact force catheters were not exclusively used in right or left AP ablation procedures. CRA catheters with 4 mm or 6 mm tips (Freezor and Freezor Xtra, Medtronic, Minneapolis, MN, USA) were used for the APs located near the conduction system at the operator’s discretion. A steerable sheath (large curve AgilisTM, Abbott, Abbott Park, IL, USA) was used for creating the ICE-guided loop needed to improve the stability between the tip of the ablation catheter and the ventricular side of the tricuspid annulus in all right-sided AP procedures [24]. On the other hand, a steerable sheath (small curve AgilisTM, Abbott, Abbott Park, IL, USA) was used only exceptionally in the left-sided procedures at the operator’s discretion.

### 2.6. Ablation of AT

Mapping of the earliest atrial activation during AT was performed with high-density mapping catheters. The location of the earliest activation was then ablated with irrigated RFA catheters, used in a temperature-controlled fashion.

### 2.7. Definition of Procedural and Follow-Up Parameters

Total procedural time (TPT) was defined as the period of time from the femoral vein puncture to the removal of the guiding sheaths. Procedural success (PS) was defined by the procedural endpoints. For AVNRT, the procedural endpoint was non-induction with or without the isoprenaline challenge. The presence of slow pathway conduction with up to one ˝echo˝ beat was allowed. Non-induction of AVRT was also tested after the successful ablation with or without isoprenaline challenge. The procedural endpoints for AVRT were the elimination of atrioventricular and ventriculo-atrial conduction over the AP. For AT, the procedural endpoint was termination of tachycardia with ablation and non-induction of tachycardia with or without isoprenaline challenge. In addition, if the source of ablation energy had to be switched to an alternative due to the inability to successfully and safely reach the endpoints, it was marked as an RFA failure or a CRA failure, depending on which energy source had to be switched. Procedures in which technical issues were the reason for the termination were excluded from analysis.

Major complications were defined as events that were directly related to the CA procedure and required an intervention, prolonged a hospital stay, and/or had a negative influence on the patient’s long-term health. Minor complications included pericardial effusion without a hemodynamic compromise, vascular complications requiring no intervention, and other adverse events that would not be qualified as major complications but were still directly related to the CA procedure. A transient high-degree atrioventricular block (AVB) that resolved during the procedure was not considered a complication.

At the follow-up visit, the patients underwent a clinical examination and had a 12-lead ECG recorded. Further diagnostic tests, such as 24 h ECG Holter monitoring with a wearable event recorder and/or repeated EP studies, were prescribed at the discretion of the physician or if the patient had signs and symptoms of recurrence. Recurrences were confirmed and noted in the EP study.

### 2.8. Statistical Analysis

Descriptive data of continuous variables were tested for a normal distribution using the Shapiro–Wilk test. Values were presented as the mean and standard deviation. Categorical variables were presented as numbers with percentages. Statistical analyses were performed using SPSS (IBM, Armonk, NY, USA) statistical software version 25.

## 3. Results

The study included 171 consecutive patients (39.8% females; mean age 12.5 ± 3.8 years; mean BMI 19.2 ± 3.6 kg/m^2^); 26.9% (46/171) of patients were younger than 10 years, while 19.3% (33/171) had a body weight below 30 kg. Multiple arrhythmias occurred in 2.3% of patients (4/171). Of the diagnosed arrhythmias, 45.1% (79/175) were AVNRT, 45.7% (80/175) were AVRT, 8.0% (14/175) were AT and 1.1% (2/175) were typical atrial flutter. Congenital heart abnormalities were noted in 5.3% (9/171) of patients, while 2.9% (5/171) of them underwent previous heart surgery. Tachycardia-induced cardiomyopathy was diagnosed in 1.8% (3/171) of patients. Baseline characteristics are listed in Table 1.

### 3.1. Procedural Characteristics

All ablation procedures were performed without the use of fluoroscopy. Overall, there were 201 procedures. ICE-guided TSP was performed in 51 (25.6%) procedures; 14 (25.1%) patients who received TSP had a BW lower than 30 kg, 11 (21.6%) had a BW equal to or below 25 kg, and only 2 had a BW was below 20 kg. RFA was performed in 79.6% (160/201), CRA in 22.4% (45/201), and a combination of both in 2% (4/201) of procedures. A total of 1.3% (2/160) of RFA procedures failed to achieve the procedural endpoints and the patients were subsequently switched to CRA. Conversely, 6.6% of CRA procedures (3/45) failed to achieve the procedural endpoints and the patients were subsequently switched to RFA. The mean TPT was 98.5 ± 55 min. PS was achieved in 99.5% of patients (200/201). There were no complications recorded in our study group during the in-hospital stay and further follow-up. Table 2 provides a summary of the procedural data.

### 3.2. Atrioventricular Nodal Reentry Tachycardia

In total, 86 CA for AVNRT procedures were performed in 79 patients. RFA was performed in 65.1% (56/86), CRA in 37.2% (32/86) and both ablation modalities in 2.3% (2/86) of the procedures. All procedures were guided by the 3D EAM system (Figure 1). Cryoenergy failed to achieve procedural endpoints in two procedures (6.3%). In the first patient, it was a redo procedure due to recurrence after the previous RFA of the slow pathway, while in the second patient, it was the first procedure. There were no RFA failures in this group of patients 0% (0/56). The overall mean TPT was 83.8 ± 51 min. PS was achieved in 100% of the procedures (86/86). The recurrence rate (RR) in this group of patients was 10.1%. The overall long-term success rate (LTSR) after the last procedure was 98.7% (78/79). There were no complications reported.

### 3.3. Atrioventricular Reentry Tachycardia

Altogether, 95 CA for AVRT procedures were performed in 80 patients. Left-sided AP, right-sided AP, septal AP (anteroseptal, mid-septal and parahisian accessory pathways were included in this group) and posteroseptal AP were present in 51.3% (41/80), 11.3% (9/80), 21.3% (17/80) and 16.3% (13/80) of patients, respectively. All left-sided and right-sided AP were ablated using RFA only. In septal AP ablation, RFA was used in 64% (16/25) of the procedures, CRA in 40% (10/25) and a combination of both techniques in 4% (1/25) of the procedures. All procedures were guided by the 3D EAM system (Figure 2). In posteroseptal AP procedures, RFA was utilized in 93.3% (14/15) and CRA in only two patients (13.3%), while both modalities were used in only one patient (6.7%). Overall, RFA failed to achieve PS in 2.4% of the procedures (2/85), and CRA in 8.3% (1/12). In one patient with parahisian AP, we started RFA with 10 W and immediately terminated AP conduction, but with a clear nearfield HIS signal recorded on the ablation catheter. Due to safety issues, we decided to switch to CRA on the same spot (marked on the 3D EAM system), which was successful in this case. Failure to achieve PS with RFA was recorded in one more patient with right lateral AP due to poor catheter stability. This patient was successfully treated in the next procedure by using ICE and the previously described loop maneuver [24]. CRA failed to achieve PS in only one patient with a low BW (22.2 kg) in an attempt to ablate ventricular insertion in the posteroseptal region of tricuspid annulus during intermittent preexcitation. The procedure was finally successfully performed with the use of RFA during ongoing orthodromic AVRT with mapping and ablation of the atrial insertion. The mean TPT was 105.2 ± 54.9 min. PS was achieved in 98.9% of the procedures (94/95), while the PS in left-sided AP was 100%. RR in this group of patients was 18.7% (15/80), or more precisely, 2.4% (1/41) in left-sided AP, 33.3% (3/9) in right free wall AP, 47.1% (8/17) in septal AP and 23.1% (3/13) in posteroseptal AP. LTSR after the last procedure was 97.5% (78/80) in all AP ablations, or more precisely, 100%, 100%, 94.1% and 92.3%, in left, right free wall, septal and posteroseptal AP ablations, respectively. There were no complications reported.

### 3.4. Atrial Tachycardia

Overall, 18 CA for focal AT procedures were performed in 14 patients. RFA was performed in 94.4% (17/18) and CRA in 5.6% (1/18) of the procedures. All procedures were guided by the 3D EAM system (Figure 3). There were no recorded RFA or CRA failures. The mean TPT was 150.6 ± 60.1 min. PS was achieved in 100% (18/18) of the procedures. RR was 28.6% (4/14). Overall, LTSR after the last procedure in this group of patients was 100% (14/14). There were no complications.

### 3.5. Follow-Up

Follow-up was completed in 100% (171/171) of patients. During the follow-up period of a mean 488.4 ± 409.5 days, 15.8% (27/171) of patients experienced recurrence after the initial ablation procedure. The mean of 1.18 procedures per patient were performed with an LTSR after the last procedure of 98.2% (168/171), and more precisely, 98.7% (78/79), 97.5% (78/80), 100% (14/14) and 100% (2/2) for AVNRT, AVRT, AT and AFL, respectively. In AP ablation, an LTSR after the last procedure of 100% was achieved in left- and right-sided AP, while one patient from the septal AP and one patient from the posteroseptal AP group experienced recurrences during the follow-up. The second procedure in these patients had not been conducted at the point of data analysis. The same applied to only one patient from the AVNRT group, in which we failed to achieve long-term success. Detailed information is given in Table 3.

## 4. Discussion

Our results show that utilizing the 3D EAM system and ICE for zero-fluoroscopy CA of SVTs in the pediatric population is feasible, effective and safe. PS rates and long-term outcomes were comparable to the published data. However, more repeated procedures were needed in septal AP ablation with CRA compared to RFA.

### 4.1. The Role of ICE in Pediatric SVT Ablation Procedures

Fluoroscopy-guided TSP is widely accepted as the conventional method for the transseptal approach in the treatment of left-sided arrhythmias in adults and children [25,26,27]. However, recently published data suggest that the addition of TOE can improve the safety of the fluoroscopy-guided TSP due to the possibility of detailed real-time visualization of the relevant anatomy [28]. Furthermore, it has been proven that ICE-guided TSP is a feasible and safe procedure in the adult population [23,29]. In comparison to TOE, ICE guidance improves the acoustic window and reduces the number of operators needed, while eliminating the need for general anesthesia and intubation, with a lower complication rate [30,31]. The slightly increased risk of vascular complication, due to a need for additional vascular puncture, can be overcome by introducing the ICE probe through the left femoral vein access in smaller children, like we practiced in our study group. Interestingly, the data on ICE-guided TSP in the pediatric population are scarce [23,32]. A retrospective study from our center published by Žižek et al. evaluated the safety of ICE-guided TSP. The study included 46 pediatric patients, and it showed an excellent safety profile. However, only seven TSPs in this study were performed in patients weighing less than 30 kg. Importantly, our current analysis included 14 procedures in children with a BW less than 30 kg, in whom ICE-guided TSPs were performed effectively and without any complications. Additionally, ICE use beyond only TSPs might bring additional safety and efficacy benefits. Friedman et al. [33] evaluated the predictors of cardiac perforation during ablation procedures in a large number of patients with atrial fibrillation. According to their analysis, ICE usage is the main modifiable factor in the current era that can improve the safety of procedures by preventing cardiac perforations. In addition, ICE can be useful as a readily available “real time” imaging method in many other situations encountered during procedures. Examples include: recognition of important anatomical variations, control of catheter tip orientation, control of catheter-tissue contact, and early recognition of complications such as pericardial effusion and procedure-related thrombi [23,31]. There are scarce data on improved results of ablation in ICE-guided AVNRT procedures [34,35]. However, we find ICE to be a valuable tool in small children with a smallish Koch triangle, when precise positioning of the catheter on the ablation spot is of great value for successful ablation and complication avoidance. In our hands, using ICE in AVNRT CRA helps avoid mechanical blockage of the slow pathway, which is the main factor for the recurrences in CRA procedures. Additionally, ICE is a very useful tool in typical atrial flutter procedures, helping the operator to overcome the anatomical obstacles that cannot be visualized adequately by the 3D EAM system or fluoroscopy [31].

Along with the use of ICE, we see the introduction of a visualizable steerable sheath as an additional valuable tool in achieving a zero-fluoroscopy CA approach. According to the recently published data, the use of 3D EAM visualizable steerable sheaths increased the number of procedures that could be performed in a zero-fluoroscopy setting, and reduced left atrial procedure time and RFA time by improving ablation catheter stability, without compromising the efficacy or safety profile of pulmonary vein isolation procedures in the adult population [36,37].

### 4.2. Catheter Ablation of AVNRT

Our results show that AVNRT in the pediatric population can be successfully and safely treated with both RFA and CRA. However, bearing in mind the risk of iatrogenic complete AVB during RFA and its consequent need for lifelong pacing in the pediatric population, we switched to using CRA as a primary energy source for slow pathway ablation. Similar to our experience, several centers have already published data on using both RFA and CRA as energy sources for treating AVNRT in children [38,39,40], while some have published results during the transition period from RFA to CRA [41]. The comparative efficacy of CRA and RFA of AVNRT in adults and children from our and others centers has already been published [18,42,43,44], with the lower RR in the CRA group (2.5% to 6.25%) in our study. We can speculate that a slightly higher RR of 10% in this study might be the consequence of the younger age and predominant RFA method of treatment. On the other hand, growing experience with cryoenergy [45], utilizing different ablation techniques [46,47,48,49,50,51], and utilizing 3D EAM systems has recently improved CRA outcomes, which are now similar to RFA outcomes [12,39,51,52]. A recent multicenter study experience from 12 centers (11 American) reported even better long-term results when using CRA in comparison to RFA [53]. Additionally, we find it important to underline that we did not experience complete AVB either in the RFA group or in the CRA group. The risk of a complete AV block as a complication of the RFA of AVNRT is the main argument supporting the use of CRA in the pediatric population, where safety is of the utmost importance. According to the PAPCA study results, the complete AVB risk in the RFA of AVNRT is 2.1% [54], while to the best of our knowledge, no study has reported complete AVB after CRA. At the same time, in accordance with the guidelines, the authors agree on the excellent safety profile of CRA, which is proven even in children with a body weight below 15 kg and especially in patients with congenital heart diseases [3,39,55,56]. In addition to the recognition of typical electrocardiograms, the approach to AVNRT ablation is also heavily dependent on anatomy. Therefore, we can speculate that using ICE in addition to the 3D EAM system could be helpful in challenging cases in which the Koch triangle might be shaped differently.

### 4.3. Catheter Ablation of Accessory Pathways

The results of our study showed that in the pediatric population with APs, both RFA and CRA procedures can be safely performed in the zero-fluoroscopy setting with an excellent PS rate, reasonable TPT and acceptable RR. As expected, the best results were recorded in the group of patients with left-sided APs, while the lowest PS and the highest RR were noted in the group of patients with septal APs.

Outcomes in left AP ablation procedures have not changed substantially in the recent decades [57], but what has changed with the introduction of the 3D EAM systems and ICE is the possibility of performing these procedures without exposing patients and the staff to the harmful effects of ionizing radiation. Using a 3D EAM system in addition to fluoroscopy has already been proven to be a factor that improves the results of AP ablation in the pediatric population [58], while utilizing ICE as a real-time imaging tool completely eliminated the need for an X-ray, even in the case of needing a left-sided approach. According to Castella et al. [59], the higher cost of the minimal-fluoroscopy or zero-fluoroscopy method due to the use of the 3D EAM system was countered by the extra costs associated with the increased cancer treatment and the reduction in the quality of life associated with conventional fluoroscopy-guided techniques. Bearing in mind the sensitivity of the pediatric population to ionizing radiation, it might even be reasonable to go beyond ALARA principles when dealing with this specific population.

Although some authors advocate retrograde transaortic ablation in the left-sided AP ablation procedures, especially in the posterolateral and posteroseptal region [60], all the left-sided procedures in our study were performed with a transseptal approach using irrigated RFA catheters with excellent PS, low RR and a satisfying long-term outcome. Additionally, regarding the procedure’s safety, we find the risk of peripheral arterial vascular complications in the younger pediatric population more probable compared to the possible complications of ICE-guided TSP for the left-sided approach.

The right free wall AP ablation procedures in our study were performed with very high PS and LTSR, which is comparable to recently published data [61]. In all right free wall AP procedures, we used real-time ICE imaging and a steerable sheath to position the ablation catheter beneath the tricuspid annulus—the so-called “loop” maneuver. This approach helped us improve catheter stability during mapping and ablation, which we found to be the main issue in achieving satisfactory results in right free wall AP ablations [24,61].

Septal AP ablation procedures are recognized as the most demanding among AP ablation procedures due to the proximity to important heart structures. Those procedures carry a non-negligible risk of complications during RFA, which includes the risk of iatrogenic complete AVB in all septal AP procedures [62] and thermal coronary artery injuries reported in posteroseptal AP procedures [63,64,65]. CRA evolved as an alternative to RFA, with promising PS rates, a significantly higher RR and an excellent safety profile and, as such, is especially attractive in pediatric electrophysiology. By utilizing both treatment options, we managed to achieve satisfactory PS and also LTSR after the last procedure, while no complications were recorded in our study group. In a recently published review of the literature and a meta-analysis of the septal AP CA data, Bravo et al. [66] compared RFA and CRA results (3495 RFA and 749 CRA), revealing a clear trend of using cryoenergy in studies that included pediatric patients. The same study showed lower efficacy of CRA compared to RFA in all septal APs (PS 86% and 89%, RR 18.1% and 9.9% and LTSR 75.9% and 88.4% in the CRA and RFA groups, respectively). This was especially true in posteroseptal procedures (PS 70.8% and 90.4% and RR 22.3% and 8.9% in the CRA and RFA groups, respectively). The only exceptions were parahisian septal APs in which CRA turned out to be a more effective treatment option compared to RFA in terms of PS (90.8% and 80.5% in CRA and RFA groups, respectively), while similarly, RR remained significantly higher (21.1% and 7.1% in CRA and RFA groups, respectively). Importantly, the complete AVB rate in the RFA group reached 2.7% in all septal pathways. The comparably low reported complete AVB rate in this meta-analysis of septal AP procedures might be a consequence of including low-risk posteroseptal AP ablation data in the analysis. When the true septal location was reported, the complete AVB risk was 7.2% in mid-septal, 5.5% in anteroseptal and 5.4% in parahisian AP RFA procedures. Our approach in posteroseptal APs is in line with the data reported in the above-mentioned study, which showed a clear advantage of RFA in comparison with CRA in this area (88.7% to 57.4% long-term success rates after multiple procedures in the CRA and RFA groups, respectively), with an acceptably low complication rate of 2.2% in the RFA group. With the use of irrigated RFA catheters, we managed to reach a favorable LTSR after the last procedure of 92.3% without any complications in this group of patients. Despite the well-known fact that RFA in the posteroseptal region can lead to coronary artery thermal injury [63,64,65], we did not record such a case in our study group.

### 4.4. Catheter Ablation of Atrial Tachycardia

A small group of patients in our study underwent an ablation procedure for AT. Not surprisingly, the PS was lower in this group—94.1%. Almost the same acute result (93%) was reported in the largest available multicenter retrospective study, which included 142 pediatric patients who underwent CA due to AT [67]. Several other groups reported significantly better results in a small series of patients [11,13]. The largest group was reported by Balli et al. [11], who achieved encouraging results by using a near-zero fluoroscopy technique with PS as high as 97.9% and a very low RR of 4.2%. In any case, data from a previously mentioned retrospective multicenter study published by Kang et al. showed that using a 3D EAM system improved the results in terms of RR (14% vs. 42%) but did not impact the PS in AT ablation procedures in children [67]. In contrast, our RR was quite high (30.8%), despite using the 3D EAM systems in all procedures. It is encouraging that additional procedures during the follow-up were clearly effective, with the final result of 100% LTSR after the last procedure.

### 4.5. Limitations

There are some obvious limitations related to our study. Firstly, weight and height parameters for a few patients were not available for analysis, which might affect the overall baseline characteristics data. Secondly, approximately 10% of procedural parameters were not available for analysis, which might affect calculated procedural outcomes; however, they were comparable to the published data, and therefore, it is prudent to speculate there is a minimal effect on the study results. Thirdly, we encountered a single patient with posteroseptal AP in whom technical issues were the reason for termination. This patient was excluded from the analysis. The result of this procedure definitely impacts the results in the posteroseptal AP group of patients, considering the relatively small number of patients in this group. Fourthly, all procedures were performed by a single operator, well experienced in using ICE and 3D EAM systems, which to some extent limits the value of our conclusions. Fifthly, we included only two patients with a typical atrial flutter in our analysis, which may mean no meaningful conclusions can be derived from the statistical analysis. However, those two patients were included to show that fluoroless catheter ablation of typical atrial flutter is feasible in pediatric patients. Finally, in our catheter ablation laboratory, we only perform fluoroless supraventricular tachycardia ablation and thus have no randomized or nonrandomized data for comparison in the pediatric population, which together with the retrospective nature of the study, limits the value of our conclusions.

## 5. Conclusions

Our analysis shows that fluoroless CA of various SVTs in pediatric patients is feasible, effective and safe. While overall PS rates and long-term outcomes were comparable to the published data, septal AP ablation procedures were somewhat less favorable with CRA compared to RFA, and repeated procedures were needed. Further studies are warranted to explore the role of ICE as a real-time imaging method in pediatric fluoroless procedures.

## Figures and Tables

**Figure 1 children-10-01513-f001:**
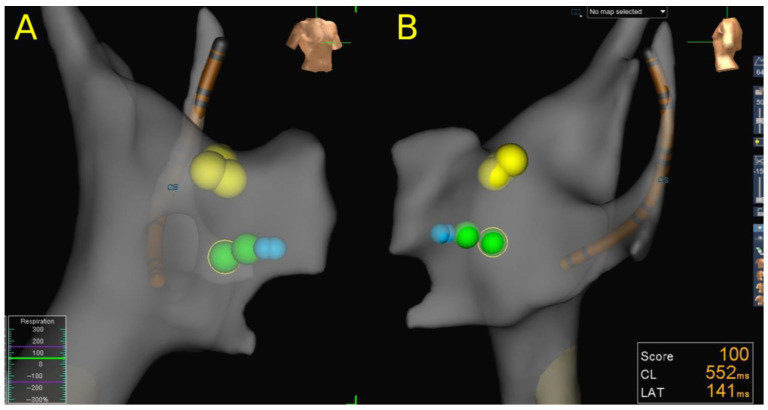
A partial 3D reconstruction of the right atrial anatomy relevant for AVNRT ablation. (**A**) Right anterior oblique view. (**B**) Left lateral view. Yellow dots mark the His location, and green dots mark the location of cryo mapping at the presumed slow pathway location. Blue dots mark CRA lesions at the initially successful cryo mapping sites. CS, coronary sinus.

**Figure 2 children-10-01513-f002:**
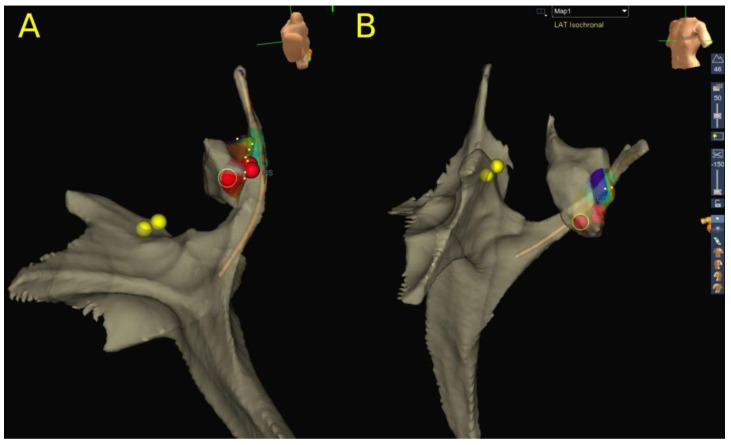
A partial 3D reconstruction of the anatomy relevant for left AP ablation with the mitral annular region activation map during preexcitation: (**A**) modified inferior view; (**B**) left anterior oblique view. Yellow dots mark His position. Red dots mark the RFA ablation lesions on the mitral annulus at the site of the earliest ventricular intracardiac signal.

**Figure 3 children-10-01513-f003:**
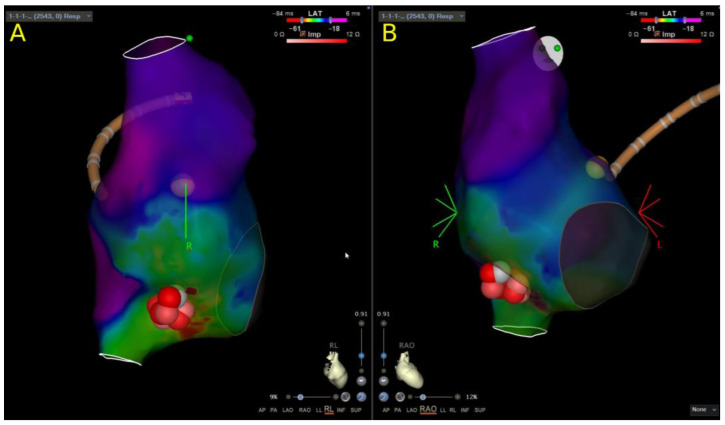
Right atrial 3D reconstruction with an activation map obtained during atrial tachycardia. (**A**) Right to left view. (**B**) Right anterior oblique view. Yellow dot marks the His location. Red dots mark the ablation lesions at the distal crista terminalis, which was the site of the earliest atrial activation during atrial tachycardia.

**Table 1 children-10-01513-t001:** Baseline characteristics.

Number of Patients	171
Female gender (Number (%))	68 (39.8%)
Age (Years) (Mean ± SD)	12.5 ± 3.8
<10 years	46 (26.9%)
≥10 years	125 (73.1%)
Weight (kg) (Mean ± SD)	49.5 ± 18.3
Height (cm) (Mean ± SD)	157.4 ± 20.6
BMI (kg/m^2^) (Mean ± SD)	19.2 ± 3.6
Prior heart surgery (Number (%))	5 (2.9%)
Congenital abnormality (Number (%))	9 (5.3%)
CIED (Number (%))	0 (0%)
Tachycardia-induced cardiomyopathy (Number (%))	3 (1.8%)
Antiarrhythmic drugs (Number (%))	35 (20.5%)
Beta blockers	19 (11.1%)
Amiodarone	1 (0.6%)
Propafenone	19 (11.1%)
Patients with multiple arrhythmias (Number (%))	4 (2.3%)
Number of all arrhythmias	175
AVNRT (Number (%))	79 (45.1%)
AVRT (Number (%))	80 (45.7%)
Right AP	9 (11.3%)
Left AP	41 (51.3%)
Septal AP	17 (21.3%)
Posteroseptal AP	13 (16.3%)
AT (Number (%))	14 (8.0%)
AF (Number (%))	2 (1.1%)

SD—standard deviation; BMI—body mass index; CIED—cardiac implantable electronic device; AVNRT—atrioventricular nodal reentrant tachycardia; AVRT—atrioventricular reentrant tachycardia; AP—accessory pathway; AT—atrial tachycardia; AF—atrial flutter.

**Table 2 children-10-01513-t002:** Procedural characteristics.

Procedural Data	All Procedures	AVNRT	AVRT	AVRT Right AP	AVRT Left AP	AVRT Septal AP	AVRT Posteroseptal AP	AT	AF
Number of procedures	201	86 (42.8%)	95 (47.3%)	13 (13.7%)	42 (44.2%)	25 (26.3%)	15 (15.8%)	18 (9%)	2 (1.0%)
Procedures per patient (Mean)	1.18	1.09	1.19	1.44	1.02	1.47	1.15	1.29	1.00
Procedure time (min) (Mean ± SD)	98.5 ± 55.0	83.8 ± 51.0	105.1 ± 54.9	142.7 ± 66.7	89.2 ± 42.3	96.4 ± 47.0	131.7 ± 66.5	150.6 ± 60.1	102.5 ± 3.5
Transseptal punctures (Number (%))	51 (25.6%)	0 (0%)	47 (49.5%)	0 (0%)	42 (100.0%)	1 (4.0%)	4 (26.7%)	4 (22.2%)	0 (0%)
Complications (Number (%))	0 (0%)	0 (0%)	0 (0%)	0 (0%)	0 (0%)	0 (0%)	0 (0%)	0 (0%)	0 (0%)
Major	0 (0%)	0 (0%)	0 (0%)	0 (0%)	0 (0%)	0 (0%)	0 (0%)	0 (0%)	0 (0%)
Minor	0 (0%)	0 (0%)	0 (0%)	0 (0%)	0 (0%)	0 (0%)	0 (0%)	0 (0%)	0 (0%)
Procedural success (Number (%))	200 (99.5%)	86 (100%)	94 (98.9%)	12 (92.3%)	42 (100.0%)	25 (100.0%)	15 (100.0%)	18 (100.0%)	2 (100%)
**Energy source used (RFA/CRA) (Number (%))**									
RFA	160 (79.6%)	56 (65.1%)	85 (89.5%)	13 (100.0%)	42 (100.0%)	16 (64.0%)	14 (93.3%)	17 (94.4%)	2 (100.0%)
RFA failure	2 (1.3%)	0 (0%)	2 (2.4%)	1 (7.7%)	0 (0%)	1 (6.3%)	0 (0%)	0 (0%)	0 (0%)
CRA	45 (22.4%)	32 (37.2%)	12 (12.6%)	0 (0%)	0 (0%)	10 (40.0%)	2 (13.3%)	1 (5.6%)	0 (0%)
CRA failure	3 (6.6%)	2 (6.3%)	1 (8.3%)	/	/	0 (0%)	1 (50.0%)	0 (0%)	/
Both used	4 (2.0%)	2 (2.3%)	2 (2.1%)	0 (0%)	0 (0%)	1 (4.0%)	1 (6.7%)	0 (0%)	0 (0%)
Number of RFA lesions (Mean ± SD)	11.6 ± 10.5	13.2 ± 13.1	11.1 ± 10.8	20.0 ± 17.0	8.95 ± 6.0	6.0 ± 3.9	8.7 ± 7.9	13.4 ± 6.4	38.0
RFA time (sec) (Mean ± SD)	374.2 ± 338.8	528.1 ± 468.8	339.7 ± 330.1	625.0 ± 502.0	275.1 ± 166.8	154.3 ± 101.4	261.3 ± 212.6	395.3 ± 129.0	935
Number of CRA lesions (Mean ± SD)	3.7 ± 2.4	4.0 ± 2.8	3.1 ± 0.9	8.0 ± 3.0	/	3.1 ± 1.1	3.0	/	/
CRA time (sec) (Mean ± SD)	881.4 ± 571.1	945.3 ± 673.7	746.7 ± 222.7	1920.0 ± 679	/	754.3 ± 256.6	720.0	/	/

AVNRT—atrioventricular nodal reentrant tachycardia; AVRT—atrioventricular reentrant tachycardia; AP—accessory pathway; AT—atrial tachycardia; AF—atrial flutter; RFA—radiofrequency ablation; CRA—cryoenergy ablation.

**Table 3 children-10-01513-t003:** Follow-up.

Follow-Up	After Successful First Procedure (per Patient)	After All Procedures (per Patient)
Follow-Up (days) (Mean ± SD)	488.4 ± 409.5	459.7 ± 391.7
Antiarrhythmic drugs (Number (%))	7 (4.1%)	5 (2.9%)
Beta blockers	7 (4.1%)	5 (2.9%)
Amiodarone	0 (0.0%)	0 (0.0%)
Propafenone	0 (0.0%)	0 (0.0%)
Long-term success (Number (%))		
All arrhythmias	144 (84.2%)	168 (98.2%)
AVNRT	71 (89.9%)	78 (98.7%)
AVRT	65 (81.3%)	78 (97.5%)
AVRT right AP	6 (66.7%)	9 (100.0%)
AVRT left AP	40 (97.6%)	41 (100.0%)
AVRT septal AP	9 (52.9%)	16 (94.1%)
AVRT posteroseptal AP	10 (76.9%)	12 (92.3%)
AT	10 (71.4%)	14 (100.0%)
AF	2 (100.0%)	2 (100.0%)

AVNRT—atrioventricular nodal reentrant tachycardia; AVRT—atrioventricular reentrant tachycardia; AP—accessory pathway; AT—atrial tachycardia; AF—atrial flutter.

## Data Availability

Underlying data may be provided by the corresponding author upon reasonable request.

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
