# Peer review of "Zero-Fluoroscopy Catheter Ablation of Supraventricular Tachycardias in the Pediatric Population"

_children, 2023, doi:10.3390/children10091513_

Round 1

Reviewer 1 Report

The manuscript by Topalovic et al. presents an investigation into the feasibility, efficacy, and safety of a zero-fluoroscopy approach to catheter ablation (CA) for pediatric patients with supraventricular tachycardias (SVTs). The study is well-illustrated, and the manuscript is well-structured; however, there are some suggestions to improve the manuscript.

In addition to the meta-analysis conducted by Yang et al. (Ref. 9), an updated and larger meta-analysis published in 2022 confirmed those findings (DOI: 10.3389/fcvm.2022.856145), which should be cited.

The methodology section adequately describes the study population and procedures, including the use of 3D electroanatomical mapping (3D EAM) and intracardiac echocardiography (ICE) for guidance during ablation. However, a potential limitation arises from the retrospective nature of the study and the absence of a control group. Without a comparison to traditional fluoroscopy-guided procedures, it is challenging to ascertain the relative advantages and disadvantages of the zero-fluoroscopy approach. Additionally, the exclusion of cases with technical issues and the limited reporting of complications might introduce bias and hinder a complete understanding of the safety profile. These points should be added to the Limitations section.

The authors state that descriptive data of continuous variables are presented as mean ± standard deviation. The t-test can be used when data distribution does not differ from a normal distribution. Did the authors confirm this? (e.g. Kolmogorov-Smirnov test).

How many operators performed the ablation procedures? What was the level of experience of these operators in using the electroanatomical mapping system and ICE?

Furthermore, the authors provided no information about whether contact-force catheters were used, which can be useful in certain clinical situations (e.g., right-sided accessory pathways). Similarly, the authors did not declare whether a steerable sheath was used during the procedures.

Importantly, the advent of visualizable steerable sheaths can further improve ablation outcomes and help achieve zero-fluoroscopy procedures, especially for operators with less experience applying the fluoroless approach (DOI: 10.1007/s10840-022-01332-8; DOI: 10.3389/fcvm.2022.1033755).

It is recommended to use the term "atrial flutter" instead of "atrial undulation".

In the Discussion section, the authors present the advantage of using ICE for transseptal puncture to avoid complications. However, using ICE can also improve procedural outcomes. Notably, ICE alone can improve procedural outcomes in patients with AVNRT (DOI: 10.1186/s12947-019-0162-2; 10.1007/s10840-022-01126-y) or atrial flutter (DOI: 10.1111/j.1540-8167.2012.02331.x.; 10.4022/jafib.1553.) compared to fluoroscopy-only or electroanatomical mapping system-guided procedures. The discussion should delve into the learning curve for ICE-guided procedures, the potential for increased procedure duration due to real-time imaging, and the necessity for proper training to avoid complications.

In conclusion, while the manuscript's aim to explore a zero-fluoroscopy approach for pediatric SVT ablation is promising, several critical aspects must be addressed. A deeper exploration of the potential benefits and drawbacks of ICE guidance would provide a balanced perspective. It is a valuable scientific work that needs improvement before publication.

Reviewer 2 Report

A very interesting, educational and informative manuscript that has clinical merit.  However, there are some editing issues that the authors should consider and address. 

The following are suggestions/comments regarding those issues. 

Line 40, "... to children due to being more ...". 

Line 188, "... modalities in 2.3% (2/86) of the procedures." 

Line 194, "100% of the procedures (86/86)." 

Line 209, "... used in 64% (16/25) of the procedures, ...". 

Line 210, "... in 4% (1/25) of the procedures." 

Line 214, "2.4% of the procedures (2/85), and ...". 

Line 219, "... poor catheter stability.  This patient was successfully ...'. 

Lines 222 & 223, "... intermittent preexcitation.  The procedure was finally ...". 

Line 225. "... achieved in 98.9% of the procedures (94/95), ...". 

Line 238, "... 5.6% (1/18) of the procedures." 

Line 240, "... in 100% (18/18) of the procedures." 

Line 334, "... the need for X-ray even in the case ...". 

Line 358, "... complete AVB [56] in all septal AP ...". 

Line 361, "and as such is especially attractive in ...'. 

Line 406, "... to the published data; and therefore, it is prudent ...".

A well written manuscript.

Round 2

Reviewer 1 Report

The manuscript has undergone significant improvements. One concern highlighted by the reviewer is the need to address a particular issue.

As an experienced operator in various types of fluoroless catheter ablation procedures, including SVT, PVI, and VT cases, I strongly believe that the introduction of a visualizable steerable sheath can greatly assist operators with limited experience in zero-fluoro procedures, enabling them to achieve minimal or even complete fluoroscopy-free ablation. These tools, alongside intracardiac echocardiography (ICE), appear to be especially valuable when a highly experienced operator, accustomed to standard techniques, attempts to transition to nearly zero-fluoroscopy procedures. For the aforementioned reasons, I find it crucial to dedicate a distinct paragraph to these tools, supported by the following references (DOI: 10.1007/s10840-022-01332-8; DOI: 10.3389/fcvm.2022.1033755), which were already included in my original review.
